# Identifying content to improve risk assessment communications within the Risk Profile: Literature reviews and focus groups with expert and non-expert stakeholders

C. Ellermann[1,2]* , M. McDowell[1,2] , C. O. Schirren[1,2], A.-K. Lindemann[3], S. Koch[3], M. Lohmann[3], M. A. Jenny[1,2,4]

**1** Harding Center for Risk Literacy, Faculty of Health Sciences Brandenburg, University of Potsdam, Potsdam, Germany, **2** Max Planck Institute for Human Development, Berlin, Germany, **3** German Federal Institute for Risk Assessment, Berlin, Germany, **4** Robert Koch Institute, Berlin, Germany

☯ These authors contributed equally to this work.
* christin.ellermann@uni-potsdam.de

**Data Availability Statement:** Most relevant data are within the paper and its Supporting information

## Abstract

### Objective

To improve consumer decision making, the results of risk assessments on food, feed, consumer products or chemicals need to be communicated not only to experts but also to non-expert audiences. The present study draws on evidence from literature reviews and focus groups with diverse stakeholders to identify content to integrate into an existing risk assessment communication (Risk Profile).

### Methods

A combination of rapid literature reviews and focus groups with experts (risk assessors (n = 15), risk managers (n = 8)), and non-experts (general public (n = 18)) were used to identify content and strategies for including information about risk assessment results in the "Risk Profile" from the German Federal Institute for Risk Assessment. Feedback from initial focus groups was used to develop communication prototypes that informed subsequent feedback rounds in an iterative process. A final prototype was validated in usability tests with experts.

### Results

Focus group feedback and suggestions from risk assessors were largely in line with findings from the literature. Risk managers and lay persons offered similar suggestions on how to improve the existing communication of risk assessment results (e.g., including more explanatory detail, reporting probabilities for individual health impairments, and specifying risks for subgroups in additional sections). Risk managers found information about quality of evidence important to communicate, whereas people from the general public found this information less relevant. Participants from lower educational backgrounds had difficulties

files. The data underlying the sub-study (focus group interviews) contain sensitive information and are protected by the Data Privacy Act. Interested researchers can submit data access requests to the ethics committee of the Max Planck Institute for Human Development (Dr. Uwe Czienskowski, sciencec@mpibberlin.mpg.de). A Codebook with details on the coding of the transcripts and number of text passages in the code per interview is available on the open science framework (https://osf.io/eqtd7/).

**Funding:** The study received financial support from the German Federal Institute for Risk Assessment (BfR) and the Max Planck Institute for Human Development (MPIB). The funding agreement ensures the authors' independence in designing the studies, interpreting the data, writing, and publishing reports. www.bfr.bund.de/en/home.html www.mpib-berlin.mpg.de/en.

**Competing interests:** The authors have declared that no competing interests exist. Risk Profiles are published by the German Federal Institute for Risk Assessment (BfR) as part of its scientific risk assessments. The publication of the Risk Profiles does not result in any economic benefits for the BfR.

understanding the purpose of risk assessments. User tests found that the final prototype was appropriate and feasible to implement by risk assessors.

## Conclusion

An iterative and evidence-based process was used to develop content to improve the communication of risk assessments to the general public while being feasible to use by risk assessors. Remaining challenges include how to communicate dose-response relationships and standardise quality of evidence ratings across disciplines.

## Introduction

To evaluate risks from food, feed, consumer products or chemicals, risk assessment agencies undertake and analyse studies to determine qualitative or quantitative estimates of these risks. Risk assessments are conducted on the basis of internationally approved scientific assessment criteria and identify potential risk groups, describe the available evidence and its quality, provide information about outcomes or define health based guidance values based on exposure to the risk (e.g., no observable adverse effect level; NOAEL), and where applicable, offer strategies for preventing or reducing the risk [1]. Risk assessments often serve as the scientific basis for decisions on the approval of products or to inform legislative actions taken by national or international bodies (e.g., risk assessment on excessive consumption of energy drinks containing caffeine by children and adolescents; [2]).

As the results of risk assessments can have implications for consumers, the effective communication of these risks to the public is essential to maintain trust in institutions, to promote appropriate perceptions of risk, and to inform consumer decision making [3]. In the medical field, a prerequisite of informed decision-making is that individuals are provided with transparent information about the benefits and harms of interventions so that they can make decisions in accordance with their preferences [4]. Similarly, consumers also require transparent information about the safety of food, feed, consumer products or chemicals, and potential mitigating behaviors or risk reduction strategies in order to make informed consumer choices.

In Germany, risk assessments are published as scientific opinions targeted mainly at expert audiences, typically risk managers (e.g., authorities at the local, state and federal level, scientific institutions), who must then communicate the information to various target groups, such as politicians, the industry and consumer associations [1]. However, many scientific opinions are freely available to the public (e.g., published online) and in some agencies it is legally mandated that they develop risk communications to inform the public about potential health risks (e.g., see [1]). At the European level, risk assessments that are considered to be of particular interest to the public are communicated in formats targeted at more general audiences, such as the European Food Safety Authority's "EFSA explains" Factsheets that develop tailored communications for specific topics (e.g., on caffeine, salmonella, or acrylamide in food; [5]). There is increasing interest in developing plain language summaries for health-related topics for lay audiences [6], similar to the approach taken in evidence-based medicine, where findings are communicated in formats targeted at the general public (e.g., Cochrane Collaboration [7]). These formats introduce a standard for reporting and communicating scientific evidence to the public and have been designed for and in collaboration with non-expert audiences [8–11].

In 2013, the German Federal Institute for Risk Assessment (Bundesinstitut für Risikobewertung, BfR) released a short graphical summary of risk assessments to enhance their

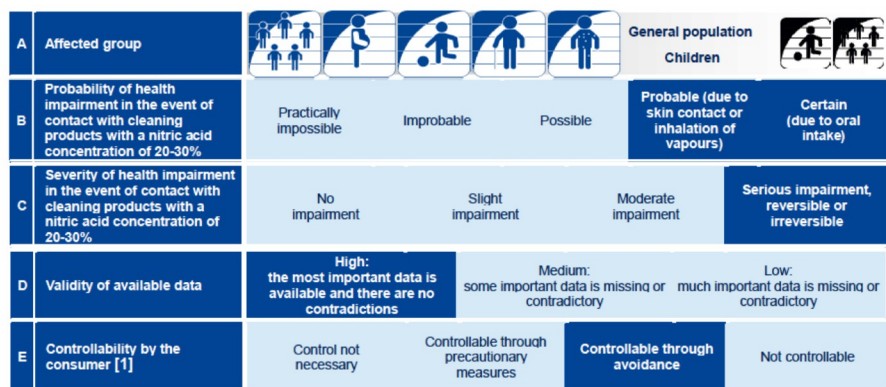

**Fig 1. Example of the 2013 Risk Profile for nitric acid in cleaning products.**

transparency and comprehensibility for both the general public and experts, a so-called BfR Risk Profile (referred to below as the 2013 Risk Profile) [12]. These summaries highlight key characteristics of risk assessments: groups of affected persons, the probability and severity of health impairments in the event of exposure, the validity of the available data and possibilities for controlling the risk through avoidance or caution (see Fig 1). For each characteristic the valid category is highlighted in dark blue (e.g., the validity of available data is rated as "High" in the example). The 2013 Risk Profile, developed to complement the more-detailed evaluations (known as Risk Opinions), are intended to represent standardised and simplified summaries of risk assessments for general audiences. Thus, the Risk Profiles present a summary of a risk assessment in a format that makes the key criteria comprehensible at a glance. The Risk Profiles are developed at the time of the risk assessment in close collaboration with risk assessors and the communications department. It was developed in a multistage process with intensive feedback, consultation and evaluation measures that took place on several levels: across departments with authors of BfR risk opinions and with representatives of external target groups of BfR risk opinions, as well as consumers.

In 2015, a qualitative evaluation of the 2013 Risk Profile was conducted to examine the comprehensibility, practical relevance and suitability of the 2013 Risk Profile for target groups. Focus group interviews were carried out with consumers, members of the BfR external expert panels, press agents, risk managers, representatives from state and federal authorities, and from business and consumer organisations. The profile was perceived to have a clear structure and provide a rough overview to encourage further interest in reading the full evaluation, but it was not perceived to reduce the thematic complexity of risk assessments. For instance, expert and professional audiences found some of the risk characteristics and their categories unclear, abstract, or contradictory. Further, experts considered the added value of the profile to be low, also reflected in the fact that few 2013 Risk Profiles were developed by risk assessors themselves to accompany risk opinions in the two years since its implementation. Non-expert audiences found the profile useful, but had difficulties understanding and correctly interpreting information. Recommendations were made to revise the profile, to include additional information (e.g., on dose-response relationships, more details on specific impairments), reduce technical terminology, and to provide more concrete recommendations for consumers and professional audiences. For instance, dose-response relationships such as Acceptable or Tolerable Daily Intake values (ADI/TDI; estimated amount of a substance in food or drinking water that can

be consumed daily over a lifetime without posing a significant risk to health) [13] have implications for consumers and risk management, but were not included in the main profile.

The objective of the present study was to improve the communication of risk assessment results by redesigning the 2013 Risk Profile, identifying content and communication strategies drawn from rapid literature reviews and focus groups with multiple expert and non-expert groups. In particular, we aimed to focus on involving diverse end-users to ensure that different information needs and preferences were adequately addressed and incorporated into the communication. Involving experts, risk assessors and risk managers in the process could help address issues of consistency, ensure that relevant risk characteristics are included, and potentially increase the adoption and use of the profile.

Consistent with recommendations for developing evidence-based health information, we aimed to recruit a diverse group of end-users to help ensure that a broad range of information needs were met [3, 14, 15]. Involving people from vulnerable groups (e.g. with a low level of education) in the development of health-related information is necessary to ensure that the resulting information provides equal opportunities for diverse groups to make health-related decisions [14, 16]. In health and social sciences, *vulnerable* refers to the fact that people of certain groups are at a higher risk of social exclusion due to factors beyond the control of the individual, such as biological factors or external environmental factors and conditions, leading to an unfair distribution of health [17]. This affects, for example, ethnic minorities, migrants, people who are disabled, ill or have a lower level of education. Further, vulnerable groups often have lower levels of health literacy and thus are more likely to have problems finding, understanding and critically assessing the quality of health-related information and applying it to their own health [18, 19]. As a consequence, information materials that are developed without involvement of individuals from vulnerable or diverse groups may inadvertently increase inequalities in health behaviour outcomes [17, 20]. For instance, designing information about consumer risks without involving individuals with lower levels of education may result in such individuals not accessing, understanding, or accepting the results of the assessment [3].

The present article describes an evidence-based process to identify and incorporate relevant information to redesign the 2013 Risk Profile to improve communication of risk assessment results. The process involved several sequential parts: a search for existing risk assessment communication strategies, rapid reviews to obtain an overview of current evidence on content and strategies to communicate information about risk assessment results into a summary "Risk Profile" (as previously designed and adopted by the German Federal Institute for Risk Assessment), and focus group studies with multiple experts and end-users to inform the development of different profile prototypes.

## Methods

In order to design a comprehensive risk profile that could be used across a variety of risk assessment topics and with different expert and non-expert groups, we used a mixed methods approach and an iterative development process. The 2013 Risk Profile design was used as a guide for the initial stages of development. In accordance with the UK Medical Research Council's guidance for developing and evaluating complex interventions [21], multiple steps were involved in revising the risk profile, with each step informing the next (see Fig 2):

(1). *Overview of existing evidence and recommendations.* An initial step involved searching for risk communication strategies or visualisation methods used by other national and international risk assessment agencies (e.g. risk matrices, pictograms), evaluate their impact on international risk and crisis communication, and identify key components to include in the communication. As this search did not yield any applicable results, we

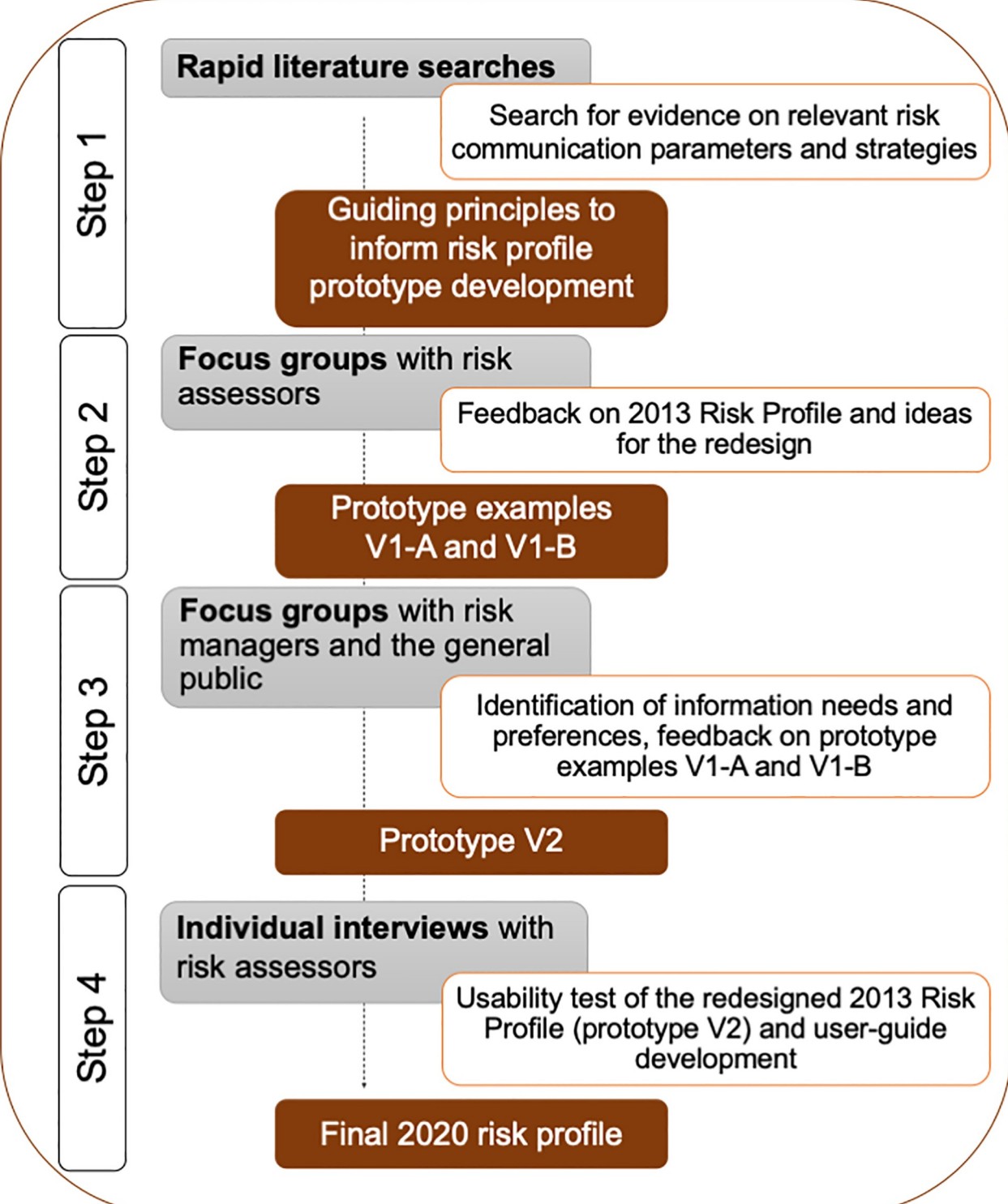

**Fig 2. Flowchart showing the successive steps taken to identify content and strategies to redesign the 2013 BfR Risk Profile.**

conducted rapid literature searches to identify any other existing formats for communicating risk assessment results and to obtain an overview of strategies for communicating specific aspects of the risk profile (e.g., how to communicate quality of evidence). On the basis of these results, a selection of categories for inclusion into risk profile prototypes were identified and guiding principles for communicating risk assessment results were summarised.

(2).   *Focus groups with risk assessors*. Focus group interviews were conducted with risk assessors to identify barriers in communicating risk assessments, to seek feedback on the 2013 Risk Profile and to generate additional design ideas or content to include in risk assessment communications. Results from (1) and (2) led to the development of several prototypes for the different content dimensions (V1).

(3).   *Focus groups with expert and non-expert user groups*. Focus groups were conducted with target audiences (risk managers, general public) to identify information needs and preferences, and evaluate different prototypes for the content dimensions.

(4).   *User-testing of prototypes*. The final prototype was validated on the basis of user-testing with risk assessors (V2). These tests also allowed the team to construct a user-guide on how to complete the risk profile.

We provide an overview of the methods for steps (1)-(4) to redesign the 2013 Risk Profile, prior to reporting the results.

## Step 1: Search and review of existing risk communication strategies

Our initial step was to search for existing tools or visualisation methods used by other national and international risk assessment agencies to communicate risk assessment results to public audiences. The aim was to identify key components relevant for risk communication that could inspire a redesign of the 2013 Risk Profile. We searched the BfR website on European and international co-operations ([22]) as well as search engines (e.g., Google) for risk assessment or risk communication institutions worldwide. In addition, experts from risk communication department of the BfR, who were involved in the 2013 Risk Profile development, were asked to share any other risk communication tools or visualization methods they were aware of. The search revealed that almost all institutions used scientific articles or reports to publish their risk assessments. Further, none used a standardized approach to communicate results of individual risk assessments to lay audiences (with some exceptions: the European Food Safety Authority produces factsheets that are non-standardized, tailored communications for specific risk assessments of public interest [5] and general information on risk assessments or food safety topics in the form of infographics [23]). The only comparable risk communication tool/visualisation method that could be identified was the Swedish National Food Agency "risk thermometer", which was developed in consultation with the BfR and is based on the 2013 Risk Profile. [24]. The risk thermometer compares food-related risks by combining probability, severity and uncertainty into a single metric that is visualized in the form of a thermometer. The purpose of the tool is to enable direct comparisons of food risks according to this combined metric and does not provide an overview of the risk assessment topic in a format that is targeted at lay audiences.

Therefore, rapid literature searches were conducted to identify and summarise different strategies for communicating risk assessment results that could inform the redesign of the Risk Profile. Like systematic reviews, rapid reviews are part of the knowledge synthesis family [25] where "processes are accelerated and methods are streamlined to complete the review more

quickly than is the case for typical systematic reviews" [26](p3). Rapid reviews are employed to provide an overview of the latest evidence on a specific topic to support decision-making while ensuring that the scientific imperative of methodological rigor is met [25, 26]. At the time we conducted the rapid reviews, there was no uniform method for conducting rapid reviews. Nevertheless, the methodological steps we took adhere to recommendations outlined in the recently published Cochrane Rapid Reviews Methods group guidelines for conducting rapid reviews (with the single exception that the protocol was not published online) [25]. We describe our methods in detail in the Supplementary Material (see S1 File).

We conducted multiple rapid reviews to summarise evidence on how to communicate: a) aspects already incorporated in the 2013 Risk Profile (e.g., probability and severity of health impairments, validity of evidence), b) additional aspects identified as being relevant to communicate about risk assessment results (e.g., dose-response relationships), and c) general risk communication practices for communicating about risk (e.g., health risks) that could be applied to communicate about risk assessment topics. Specifically, we reviewed literature on communicating the following key questions:

- How to communicate information about food risks or health risks from consumer products in general,

- How to communicate probabilistic information (e.g., quantifiable or quantitative information),

- How to communicate information about the severity of health impairments,

- How to communicate uncertainty related to the quality of the underlying evidence, and

- How to communicate dose-response relationships or thresholds.

Several search queries for each research question were developed by two researchers (CE, CS) and supported by an information specialist from the library of the Max Planck Institute for Human Development. The searches were conducted in the scientific databases PubMed, PsycINFO, Scopus and Web of Science by one of the lead authors (CE). Owing to the different content within each topic, the use of databases were adjusted to the specific search such that the same set of databases were not used for all key questions. Specifically, for some searches, the use of additional databases were employed to find relevant studies (e.g., the use of educational research databases to search for dose-response relationships). Relevant studies were identified according to predefined selection criteria and were evaluated and summarised narratively. To prioritise the most recent evidence, we focused the search on peer-reviewed systematic reviews and randomized controlled experiments published in English or German in the last ten years. Preference was given to systematic reviews (with or without meta-analyses), with RCTs supporting or supplementing these reviews where they were outdated or where no systematic reviews were available. For the question on how to communicate probabilistic information, the search was limited to systematic reviews published within the last five years given that the field is already well-researched [27–29]. Details about the literature search are available in a short report in the supplementary materials (see S1 File). We provide a brief summary of the results in the results section to illustrate how they informed subsequent stages of the risk profile revision.

## Step 2: Focus group interviews with risk assessors

Focus groups with risk assessors focused on identifying core content or assessment characteristics that were most relevant for risk communications, to elicit feedback on the 2013 Risk

Profile and to inform the development of risk profile prototypes. The reporting follows the Standards for Reporting Qualitative Research (SRQR) [30] (see S2 File). Two focus group interviews were conducted in June 2018 (n1 = 7, n2 = 8) by two moderators (CE, CS) with different scientific backgrounds and a member of the BfR research team (ML) at the BfR premises. Both moderators had no previous experience in moderating focus group interviews, but were fully briefed by the team's supervisors. Each focus group lasted two hours. The participants from all scientific departments at the BfR were invited to participate on a voluntary basis. The focus groups included participants with and without previous experience working with the 2013 Risk Profile. Participants were primarily risk assessors from various departments of the BfR (n = 13), and included members of the press and legal units who were involved in communicating about risk assessments or the legal aspects of BfR risk communications (n = 2). Due to the small number of participants, differentiated information on the fields of work and demographics is not reported in order to ensure data protection. Participation was voluntary and focus groups were conducted within the participant's normal working hours. Ethical approval was obtained from the Max Planck Institute for Human Development Ethics Committee. Written declaration of consent to participate in the study and for audio recording was obtained from all participants before the focus groups' interviews commenced. Participant statements were anonymised during transcription. Personal contact data and transcription data are stored separately and treated as strictly confidential.

The focus groups followed a semi-structured question guide (see S3 File). After presentation of the 2013 Risk Profile participants were asked for their opinions on the profile (see Fig 1) and its individual characteristics. Second, suggestions were sought on how to improve the content or communication of results within each of the five characteristics, and for ideas on visual designs. Interviews were audio recorded and field notes were taken. Interviews were transcribed with the software *f4transkript* by a student assistant (MM) and checked for completeness and accuracy by one of the lead authors (CE), and subsequently analysed using *f4analysis*. The Kuckartz methodology [31] was used for qualitative content analysis. On the basis of question guides, a deductive coding system was developed and supplemented with an inductively developed coding system. Coding and summary of the interviews was done by one of the lead authors (CE). Quotes were added to the summaries of the respective interviews. The second moderator who was also present during the interview, reviewed, revised, and discussed the summaries of the findings based on her field notes. The second lead author (MMD) and researchers from BfR research team (ML, SK), who were not involved in conducting and evaluating the interviews also reviewed the results. The translation of the quotes presented in this article from German into English was done by the lead authors and a student assistant (CE, MMD, MM). Consensus was reached through discussion. The codesystem and codebook are available on the Open Science Framework (OSF) (https://osf.io/eqtd7/).

## Step 3: Focus group interviews with target user groups

Semi-structured focus group interviews were conducted with risk managers and members of the general public to elicit feedback on risk profile prototypes developed on the basis of Step 1 (literature search) and Step 2 (focus group interviews with risk assessors), and to inform further revisions of the prototypes prior to user testing (step 4) (see S3 File). Questions focused on comprehensibility, acceptance and applicability of the prototype examples, preferences for design, and suggestions for improvement (e.g., "In your opinion, what would be the best way to describe the probability of a risk occurring?" or "How do you think the severity of the health impairment can best be communicated?"). Participants (n = 8) from the field of risk management were recruited from German ministries and federal offices by the BfR. Three men and

five women were recruited, with five participants from the BVL (Federal Office of Consumer Protection and Food Safety) and three participants from the BMEL (Federal Ministry of Food and Agriculture). Risk managers from these agencies use the results of risk assessments from the BfR to make decisions about risk management for consumers (e.g., determine limit values for certain substances in food; withdraw products from the market). Due to the small number of participants more differentiated information on the fields of work and demographics is not reported in order to ensure data protection.

Two focus groups were conducted with participants from the general public. The participants were recruited by an external market research company with the goal to recruit individuals from various backgrounds (e.g., age, education). Including individuals from diverse backgrounds helped us to address potential differences in information needs, as well as comprehension and use of risk assessment information. In the first group (n = 8), the majority of participants had completed vocational training (n = 5) and their mother tongue was German (n = 7). The mean age was 44 years (age range 26–58 years) with equal gender representation. In the second group (n = 8), three participants had completed vocational training, four had completed their studies (master or bachelor), one participant was currently studying and most reported German as their mother tongue (n = 7). The mean age was 43 years (age range 23–62 years) and two participants were women.

Focus groups were conducted in December 2018, with each group lasting 90–120 minutes. Ethical approval was obtained from the Max Planck Institute for Human Development Ethics Committee. Written declaration of consent to participate in the study and for audio recording was obtained from all participants before the focus groups' interviews commenced. Participant statements were anonymised during transcription. Personal contact data and transcription data are stored separately and treated as strictly confidential. The participants from the general public each received 30 Euro as remuneration for their time. Risk managers did not receive remuneration for participation; the interviews took place within their working hours.

Semi-structured interview guides were developed for both target groups. The focus group interviews were conducted by one researcher from the HC (CE or CS) and accompanied by at least one other researcher from the research team (AKL, CS) at the premises of a market research company in Berlin. Both moderators were involved in the previous focus group interviews with risk assessors and subsequent development of the risk profile prototypes. The interviews were audio-recorded and field notes were taken. Focus group interviews with risk managers and people from the general population were transcribed by an external market research company. The transcripts were double checked by one of the lead authors (CE) and a research assistant (MM). Also here, in accordance with Kuckartz [31] a deductive procedure was used in which the codes were defined beforehand on the basis of the respective question guide and applied to the text in order to guarantee the validity and reliability of different coders. In a second step, the category system was further developed during coding on the basis of the text (inductive method). The qualitative data analysis software f4analyse *f4analysis* was used again for data analysis. The coding and summary of the interviews was done by one of the lead authors (CE). One additional interview was coded by a research assistant (MM) who was not involved in the development of the code system and in conducting the interviews. The codesystem and codebook are available on the Open Science Framework (OSF) (https://osf.io/eqtd7/). Quotes were added to the summaries of the respective interviews. The researchers on the research team, who were also present during the interviews (CS, AKL), reviewed, revised, and discussed the summaries of the findings based on their field notes. The second lead author (MMD), who was not involved in conducting and evaluating the interviews, also reviewed the findings.

## Step 4: Usability test and guide development

Following focus group interviews with risk managers and the general public, usability tests were conducted to elicit additional feedback from risk assessors on completing the profile and to validate a risk profile prototype V2 developed after step 3 (focus group interviews with target user groups) and to develop a user guide. Two members of the research team (CS, AKL) who had previously been involved in conducting or accompanying the focus groups met with authors (n = 5) of three existing BfR risk opinions at the BfR premises. The usability tests were conducted in March and April 2019 and took one to two hours each. The sessions took place during standard working hours. No further declaration of consent was necessary, as the participants had already agreed to be contacted again during the focus group interviews. Risk assessment topics were selected across multiple BfR departments to address different risk communication challenges across disciplines: different level of familiarity among consumers with the issues as well as opinions that did and did not contain a health based guidance value. The selected topics were magnesium as a food supplement, brucellose in horse milk, and pyrrolizidine alkaloids in plant-based food and food products.

An open approach was used for the usability tests. Risk assessors received a printed copy of the original text-based summary and 2013 Risk Profile, an empty template of the revised risk profile (prototype V2) and a draft user guide on how to complete the profile. The user guide was drafted based on the V2 prototype and aimed to provide risk assessors with guidance on how to complete the risk profile (e.g., what information to provide in each section; best practices for presenting numerical risks etc). It was developed prior to the usability tests to assist risk assessors in completing a Risk Profile based on one of their recent risk assessment topics, and would be revised based on feedback from risk assessors during the usability tests. As a first step, risk assessors and members of the research team completed the redesigned risk profile together on paper, and any implementation issues were identified and discussed with the team members. Following the session, the hand-written risk profiles were transferred to digital documents and sent to risk assessors for confirmation. Multiple feedback rounds occurred until the respective risk profiles were completed.

Based on the input from these different methods and analysis, the prototype (V2) of the revised risk profile was validated and a final 2020 Risk Profile developed.

## Results

### Step 1: Literature reviews

Across all conducted searches, a total of 19 studies were identified that evaluated the effectiveness of specific strategies (n = 9 reported on communicating probabilities, n = 7 on communicating quality of evidence, and n = 3 on communicating severity of the impairment) and three studies reported on best practice recommendations for communicating food or health risks in general. Crucially, we found no prior studies or systematic reviews on experiments to communicate the results of risk assessments to the general public. Further, for most of the five risk characteristics, there were no high quality studies (randomized controlled trials) or reviews for communicating results about the severity of health impairments, dose-response thresholds, or the uncertainty or quality of evidence. Only on the question on how to communicate probabilities was a substantial amount of high quality evidence found.

A summary of the literature review methods and results can be found in the Supplementary Material (see S1 File; a detailed report on the literature reviews is available on request). For brevity, Table 1 provides an overview of guiding principles for developing risk assessment communications derived from the literature reviews.

**Table 1. Summary of guiding principles for developing risk assessment communications derived from the literature review.**

**Guiding Principles for Risk Assessment Communications**

*Profile development process*

- Involve different target groups and adapt prototype designs to focus on lay understanding.

*Presentation and communication of risk assessment criteria*

- Adapt the layout, structure, and design of texts to ensure readability for less educated groups (e.g., using a question-answer format, clear structure of texts, use of examples, color design; [15]).
- Where possible, quantify or categorize risks and use numbers to communicate the magnitude of risks. A recommendation against using verbal labels unless accompanied by numbers was advised, as verbal labels are interpreted inconsistently [15, 28, 29, 32].
- Define a clear reference class when reporting numbers (e.g., frequencies or percentages; [15, 27, 29, 32]).
- Provide probability and severity information separately for each concrete health impairment.
- Consistent with recommendations on communicating scientific uncertainty, include the quality of evidence despite few studies on how best to represent this information. As a first step, communicate non-quantifiable uncertainty as a predefined categorization of uncertainty (e.g., low, medium, high), otherwise as a qualifying verbal statement based on an evaluation of aspects including study limitations, consistency of results, indirectness of evidence, imprecision, and reporting bias [33, 34].
- Add a description of specific actions that would reduce the risk for consumers.

*Criteria lacking evidence base*

- Dose-response thresholds: no studies on how to best communicate these could be identified. Ideas were generated within the research team (e.g., depicting a thermometer, a curve, colour gradients, etc.).
- Use of analogies for communicating about complex concepts

## Step 2: Focus group interviews with risk assessors

Risk assessors and members of the communication and legal department identified a number of challenges they face when preparing risk assessments and communicating the results to broader audiences. For instance, participants mentioned that they often cannot provide concrete probability information (numbers) because data are not available, especially for specific risk areas or for risk groups (the so-called "YOPIs": young, old, pregnant, immunosuppressed). For example, for some risk assessments, risk information is based on individual cases where people have been accidentally exposed to a substance or have ingested something improperly (e.g., a child drinking detergent), or on an expected risk calculated on the basis of a single outlier (e.g., a limit value has been exceeded or a contamination has only been found in a few cases). In other cases, risk information may not be available on humans or there may be no information available because studies cannot be conducted for ethical reasons. Further, health impairments depend on the intensity and duration of exposure (dose-response), and risk assessors stated that these relationships cannot be adequately expressed in the current probability rating format. Risk assessors stated that it is important to include more specific background information about the evidence on which the risk assessment is based.

Participants made suggestions on how to address some of these challenges or to improve on the current risk profile, typically by providing more flexibility or detail to the profile. For instance, they suggested to show only the relevant categories for each risk characteristic (e.g., to say only "no impairment" in the severity section) and to provide some explanations instead of showing all possible categories (of which most may not apply) and highlighting the relevant one. Further, they also suggested using questions to describe each dimension to non-expert audiences, and to include more differentiated information for different risk groups (e.g., separate lines or columns for children and pregnant women). Risk assessors also thought it would be important to list specific endpoints instead of just stating that the severity of a health impairment is low, medium or high:

*[. . .] if you read this as a consumer: What does 'serious impairment' mean? What is happening?*

[INT2-Risk Assessor]

Risk assessors also thought that it was important to communicate probability data more transparently, as only providing verbal terms was unclear, and to describe the quality of the data more precisely. One participant suggested that the risk profile should be used to better structure risk assessments so that the main text and risk profile were better aligned. Risk assessors could then fill out the risk profile themselves, taking into account information for different groups, but could also state when information is not available for a specific subgroup (e.g., children).

*Maybe [use it as] some kind of risk profile combined with a checkbox* [checklist] *So, here 'on children': "Yes, there is data", 'on pregnant women': "No, there is no data". Or 'for children': "is relevant"*

[INT1-Risk Assessor]

An additional challenge relating to the controllability dimension that was mentioned by participants was that German Federal Law mandates a strict separation between risk assessment and risk management to maintain independence. Specifically, while risk assessment institutes are allowed to give recommendations on risk management, their implementation is solely the responsibility of risk managers. However, risk assessors mentioned that recommendations are often a logical conclusion of risk assessments (e.g., to avoid preparation and consumption of non-heated cocoa drinks or other milkshakes made from raw milk). In addition, for some assessments, it is not in the hands of consumers to control the risk themselves (e.g., when plant-based foods are contaminated with inedible weeds), which could lead to frustration and resignation, which is why manufacturers and policy makers should also be addressed.

To summarise, on the basis of focus group feedback, suggestions on how to improve the communication of risk assessment results in a risk profile are presented in Table 2.

The focus group results were integrated with suggestions from the literature reviews to create two different prototypes (V1-A and V1-B) for individual dimensions of the risk profile to test in subsequent developmental stages (see Fig 2). Fig 3 provides an example of the probability and severity dimensions for the two prototypes using the example for the risk assessment on raw milk consumption [35]. Prototype V1-A was similar to the original 2013 Risk Profile

**Table 2. Summary of suggestions on how to improve the 2013 Risk Profile from focus groups with risk assessors.**

- Include a section introducing the problem addressed and the context for the risk assessment at the beginning of the profile.
- Use a question-and-answer format for each dimension.
- Include more visualisations, where possible.
- Provide a more comprehensive summary and link to the respective chapter in the risk assessment.
- Add a dimension about exposure context (e.g., inhalation, skin contact).
- Include a section for communicating thresholds (e.g., a limit value), either as a separate section (where relevant) or in place of probability and severity.
- Communicate quality of evidence about toxicity and exposure separately
- Address consumers as well as manufacturers, risk managers, or governments in a section on how to control risk.
- Structure the dimensions and/or profile such that additional lines can be included in each section for subgroups (e.g., separate lines for affected groups, different exposure contexts) or, if outcomes differ substantially across affected groups, create complete separate profiles for each subgroup.

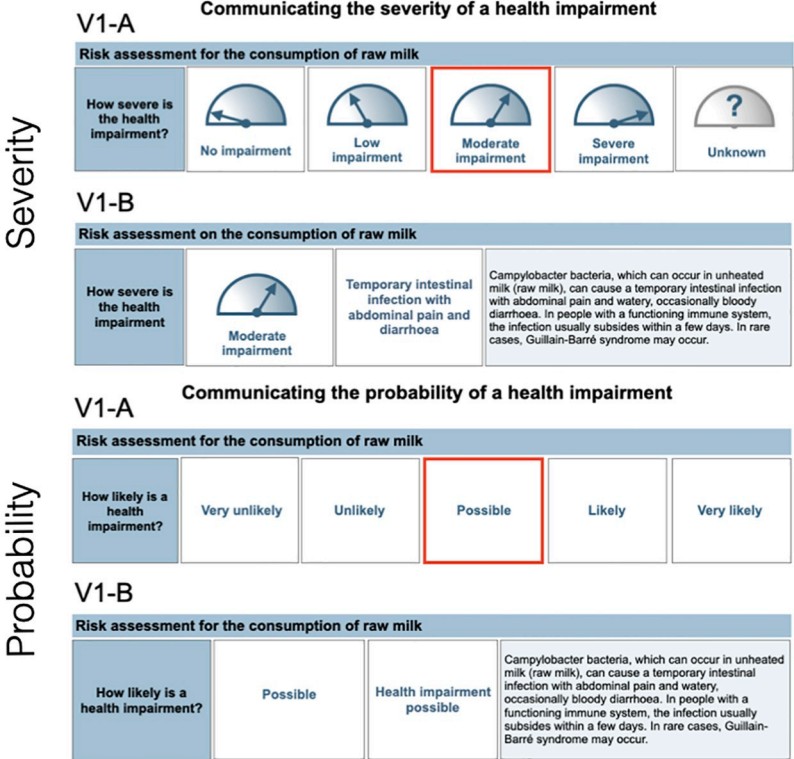

**Fig 3. Prototype examples of V1-A and V1-B developed to communicate the severity and probability of outcomes dimensions.**

and contained all possible categories (e.g., all levels of severity or the probability of a health impairment) and highlighted the applicable one. Prototype V1-B only showed the applicable category with additional detail provided as a short explanation.

We also developed a prototype to communicate health-based guidance values, such as the Acceptable or Tolerable Daily Intake (ADI, TDI), or health-based guidance value (see Fig 4). As the literature search identified no existing visualisation tools to communicate threshold values, we developed multiple designs based on general principles for visual design (e.g., culturally appropriate use of colours, use of transparency or blurring for communicating uncertainty, including numerical values) and team discussions.

## Step 3: Focus group interviews with risk managers and people from the general public

There were many similarities in the feedback received from risk managers and members of the general public. For instance, almost all participants preferred prototype V1-B, which only showed relevant categories and contained more detailed information (see Fig 3):

> *Then it's perfectly clear that you leave everything else out. What do I need the other information for, if it's not applicable anyway?*

[INT5-General public]

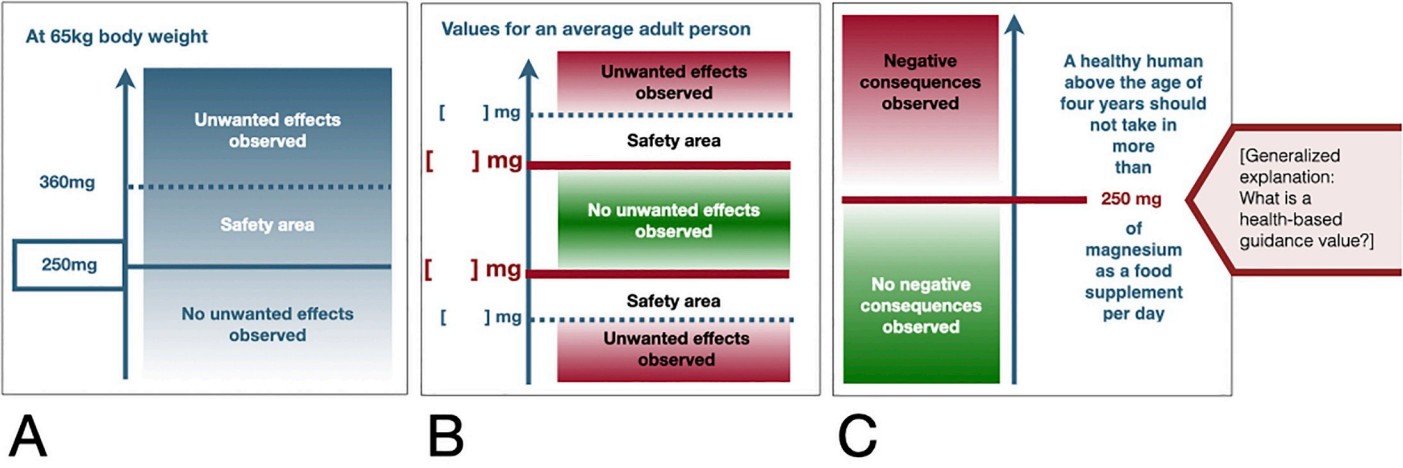

**Fig 4. Visual designs for communicating the health-based guidance value as an example.** Designs were developed and modified during different stages of the project. (A) The initial design discussed in focus groups with risk managers and lay people. (B) Design for cases where both upper and lower threshold values were possible. (C) The final version developed for the magnesium as a food supplement topic after usability tests with risk assessors.

Further, participants wanted information about subgroups to be differentiated (e.g., more sensitive groups, like children) and asked for precise definitions for the verbal probability terms (e.g., ideally, using numbers). Only with regard to the certainty of data and the controllability of risks were there differences between the two audiences. Non-experts found information on the certainty of data too scientific and not relevant for themselves, whereas risk managers found the information relevant, but thought that it may be too complicated for the lay public:

*It's going too far* [certainty of data]. *There's already too much written on the packages anyway. The purpose of this institute is to warn people about certain dangers. You have to be able to spot them quickly. A colourful icon that everyone can read and understand immediately, and if you write down all kinds of data and descriptions, it won't lead to the goal. [. . .]*

[INT4-General public]

*I'm already pretty much into the scientific area. I don't know whether there are so many people who are really interested in this and whether anyone can do anything with this information, that it should be made available to everyone, but that it should perhaps be shown under 'Further information'.*

[INT5-General public]

*I honestly believe that the consumer is not so much interested in what data is behind it, what quality, etc., but that they actually only want to know what is the best possible option for action for them personally [. . .]. What this information is based on and how we as risk managers produce it is, I believe, relatively unimportant to them. And that's why this is now a very decisive question for me. In my view, the full range of criteria currently contained in this risk profile is, in fact, perhaps more aimed at us. For the consumer I find it too diverse.*

[INT3-Risk manager]

Regarding the controllability of risks, risk managers wanted to receive clear recommendations for how to control risks based on the risk assessments, whereas people from the general public found the information in the given example (consumption of raw milk) to be self-explanatory.

Non-experts also had problems understanding the general purpose of risk assessments. For instance, it was unclear to them why risk assessments were conducted when there was a high level of uncertainty in the data or why products were available for consumers when there is a risk of health impairment. It was also challenging for them to think beyond the specific content example used in the focus group. In contrast to non-experts, risk managers did not find the visualisation of thresholds easy to understand for the general public and requested more concrete explanations of threshold values.

To summarise, on the basis of feedback from the focus groups, the changes that were made to develop prototype V2 (similar to final prototype of the 2020 risk profile in Fig 5; minor wording modifications were made on the basis of user-testing, as described below) are presented in Table 3.

## Step 4: User testing of prototype V2 and user guide development

Risk assessors perceived the prototype V2 as an improvement on the 2013 Risk Profile. In particular, risk assessors liked the layout, structure, use of simple visuals for each category, and the possibility to include a short text to further explain each dimension. Some of the risk assessors thought the updated risk profile had sufficient detail and structure to replace the standard summary text. However, this opinion was not shared by all risk assessors. Regarding the implementation of the risk profile, some risk assessors suggested to link the categories in the risk profile to the respective sections in the risk assessment document to allow readers to directly jump to sections of interest.

Multiple designs for the visualisation of a health-based guidance value for communicating dose-response relationship were discussed with risk assessors, as an appropriate strategy had not yet been decided on based on feedback from prior developmental stages. Risk assessors suggested to communicate dose-response relationships using a green and red colour grading to depict dosages with a positive and a negative health effect, respectively. To reduce confusion resulting from trying to communicate several threshold values (e.g., the no observable adverse effect level (NOAEL), the lowest observable effect level (LOAEL) and a health-based guidance value), risk assessors suggested that only the most relevant value should be communicated in the risk profile. For instance, in the example for magnesium as a food supplement, only the health-based guidance value is shown (Fig 5), because it was considered to be more relevant to consumers than other health-based guidance values, such as NOAEL or LOAEL.

Quality of evidence was assessed using different, discipline-specific conventions. While some risk assessors suggested to use the GRADE system [34], a well-established system for assessing quality of evidence in evidence-based medicine, others stated that this would not be appropriate within their disciplines. Resolving discrepancies between the conventions used across different scientific disciplines was beyond the scope of the present study. Nevertheless, to promote transparency, it was recommended to provide a description of the type of studies underlying risk assessment results to explain why a specific level of quality of evidence was chosen (e.g., study design, whether studies included many or few participants, and the consistency of studies). Risk assessors also thought that such explanations could have the added benefit of helping to inform consumers of the quality associated with different sources of scientific evidence.

| | | |
|---|---|---|
| **- Risk opinion -**<br>**Too much magnesium as a food supplement can lead to diarrhea** | | |
| **In a nutshell:** | **Too much magnesium, e.g., taken as a food supplement in addition to magnesium intake through a normal diet, can lead to diarrhea** | • Magnesium is an essential mineral. Some people take magnesium in addition to their normal diet in the form of **food supplements**. If too much magnesium in the form of a food supplement is consumed, temporary diarrhea may occur.<br><br>• The risk can be reduced by not taking magnesium-containing food supplements. If magnesium is taken in food supplements, a daily intake of **250 mg should not be exceeded**. This opinion applies to healthy adults and adolescents, including children aged four years and older. |
| **Who is affected?** | **People who take magnesium as a food supplement** | The opinion applies to healthy adults and adolescents, including children from the age of four. For infants under four years of age, no maximum level can be determined due to a lack of data. |
| **How is magnesium as a food supplement consumed?** | **Oral intake** — **Through consumption of magnesium-containing food supplements** | The intake takes place through the consumption of magnesium-containing food supplements - usually in the form of effervescent tablets or powder. |
| **How much magnesium as a food supplement can I take?** | **Negative effects observed** / **No negative effects observed** — **A healthy person from the age of four should not consume more than 250 mg of magnesium as a food supplement per day** | Taking **more than 250 mg of magnesium** in the form of food supplements per day increases the risk of **temporary diarrhea**. The more magnesium is consumed, the higher the risk of diarrhea.<br><br>The **health-based guidance value** is a value that is below the dose at which adverse effects of a substance have been observed in humans, animals or cells. For values below the health-based guidance value, adverse effects of a substance are **unlikely**, but can never be completely excluded in individual cases.<br><br>Negative effects have **not been observed** in humans for intakes of 250 mg or less of magnesium per day through food supplements. When taking magnesium in food supplements, check how much magnesium is contained by reading the **package label**. |
| **What is the quality of evidence?** | **High** — **On health risks from magnesium in food supplements** | Regarding health risks (temporary diarrhea) caused by magnesium as a food supplement, a number of scientific studies have been carried out, including with humans. The number of participants included in the studies was small compared to other studies in this field, but they were of high quality and the results were consistent. |
| **How can the health risk from too much magnesium through food supplements be reduced?** | **Government** — **Determination of maximum consumption levels for magnesium in food supplements** | The Government can reduce the risk from magnesium in food supplements by setting maximum levels for magnesium in food supplements. In doing so, it should be considered that the total daily intake of magnesium from food supplements should not exceed 250 mg. |
| | **Producers** — **Adhering to recommendations for maximum consumption levels in food supplements** | The industry can reduce the risk from magnesium in food supplements by making sure that the recommended maximum level of 250 mg for a daily intake is not exceeded in consumption recommendations for food supplements. Consumers can be informed through package labels. |
| | **Consumers** — **Avoiding magnesium-containing food supplements** | Consumers can reduce the risk from magnesium in food supplements by avoiding magnesium-containing food supplements. Even under adherence to the maximum daily level, the BfR recommends to split the total dosage into at least two intakes per day. |

**Fig 5. Final version of the 2020 Risk Profile on magnesium as a food supplement, as developed after usability testing.**

**Table 3. Summary of changes to the initial prototype examples (V1) from focus groups with risk managers and general public.**

- The presentation of outcomes was based on Prototype V1-B, as this was preferred almost universally by participants from all focus groups and matched the suggestions made by risk assessors.
- The severity and probability dimensions were combined, as participants felt outcomes were better understood when presented together, and different outcomes occurred with different likelihoods. Further, as severity-ratings (e.g., mild, moderate, or severe) were perceived as too vague, each health impairment was specified in concrete terms.
- When quantification of the likelihood of occurrence for a health impairment was not possible, a comprehensive explanation regarding what is known was provided and a relevant verbal category highlighted (e.g., likely, unlikely). However, where quantification is possible, numbers should be provided.
- When threshold values, such as a health-based guidance value, were communicated, we communicated them in the section referring to severity and probability, accompanied by an explanation and depicted using a visual aid.
- Despite some non-expert participants finding quality of evidence information unnecessary, it was retained in the profile. Communicating this type of information is currently recommended to accurately reflect scientific knowledge and increase public understanding and tolerance towards uncertainty [36]. Risk assessors also found it to be important to communicate this information. To simplify the presentation, it was decided that only three categories should be used to describe the quality of evidence—"Low", "Moderate" and "High"—and for an explanation to be provided.

Based on the usability tests, the modifications made to the prototype V2 are presented in Table 4. Fig 5 presents the redesigned Risk Profile (2020 Risk Profile) for the topic of magnesium as a food supplement based on the 2013 Risk Profile. Further, the draft user guide was developed in collaboration with risk assessors during the usability test. The purpose of the user guide was to provide risk managers with detailed instructions on how to complete each section of the risk profile, including guidance on what information to include and recommendations about what formats to present information to facilitate interpretation (e.g., to include numerical probabilities, if available, in absolute rather than relative numbers). These recommendations were also informed based on findings from the literature reviews (e.g., see Table 1). The draft user guide is currently being revised revised in coordination with working groups within the BfR.

## Discussion

The aim of the project was to identify content and strategies to incorporate into an existing standardised format to communicate the results of scientific risk assessments to the general public. Using an iterative mixed-methods approach, we redesigned the 2013 BfR Risk Profile, taking into consideration the information needs and preferences of diverse end-users (risk

**Table 4. Summary of modifications to the prototype V2 from usability tests.**

- The term "health-based guidance value" was used to describe an upper or lower limit of exposure to or consumption of a substance. Risk assessors stated that this term described relevant threshold values, such as the Acceptable and the Tolerable Daily Intake (ADI and TDI), in non-expert terms. Where an upper and lower limit of exposure was relevant (e.g., consuming an amount of fish to increase health benefits while limiting mercury exposure), both limits should be described, similar to Fig 4B.
- The safety area was removed from the health-based guidance value visual as risk assessors stated that the size of a possible safety area could not be measured. Rather, the difference between the no observable adverse effect level and the upper limit of exposure should be communicated as an uncertainty interval (see Figs 4 and 5)
- For quality of evidence, details about the type or source of available evidence (e.g. animal studies, expert opinions, cell studies etc) were described.
- Minor edits were made to the wording of different categories (e.g., the controllability dimension "How can the risk be minimized" was changed to "How can the risk be reduced").

managers, people from the general public). We took care to also recruit individuals from more disadvantaged groups (e.g., lower education) who typically benefit less from broad-based interventions which in turn increases health inequalities [20]. We also involved risk assessors, who are also the authors of risk assessments, in multiple phases of the development cycle. Thus, we were able to gain valuable insights into the challenges they face when summarizing risk assessment results, ensure that all relevant characteristics were included in the format, and to engage them in the redesign of the 2013 BfR Risk Profile in accordance with their needs.

Even though we sought feedback from a variety of different stakeholders, many suggestions were shared by members across all user-groups. For instance, people from the public and risk managers had largely overlapping preferences and came up with similar suggestions on how to improve the communication of different elements—suggestions that often matched those made by risk assessors or that were identified in the literature searches (e.g., specifying concrete endpoints for health impairments, using concrete numbers instead of probability statements, showing only relevant categories). The project also identified a number of open questions and ongoing challenges related to the communication of risk assessment results. For example, a major finding of our literature searches was the clear absence of evidence (e.g., relevant and/or high quality studies) on risk communication strategies for certain risk characteristics. We discuss some of these remaining challenges below.

## Quantifying probabilities in risk assessments

A clear recommendation for communicating probabilities is that numbers should be used, as verbal labels alone (e.g, "common", "uncommon", or "rare") are interpreted inconsistently [15, 28, 29, 37, 38]. Indeed, although we did not include a risk assessment topic that presented risks numerically, there is an extensive literature on how to present probabilities in formats that facilitate understanding (e.g., absolute numbers rather than relative risks; including information on base rates or reference groups; [39]). However, risk assessments often do not contain the type of data necessary for quantifying probabilities related to health outcomes. For instance, randomised controlled trials on substances that have been found to be toxic in cell cultures or in animal studies cannot be conducted on human participants. Consequently, it is not possible to estimate the probability that a human being would experience a health impairment given different levels of exposure. In such cases, risk assessments often rely on data from individual human cases or animal studies that are difficult to extrapolate to meaningful risk magnitudes for humans. In the redesigned risk profile, we attempted to address this challenge by providing contextual information about the studies the evidence was based on (e.g. "multiple case studies with humans found damage to liver and lungs") rather than using a verbal probability (e.g., "damage to liver and lungs is possible"). Further, specifications and more detailed explanations for the preparation and presentation in the BfR opinions can be found in the new edition of the BfR guidelines for the evaluation of health risks [40]. However, further research is needed to understand how to communicate non-quantifiable risks to non-experts.

## Criteria for rating quality of evidence across scientific disciplines

In health research, a lot of effort has been dedicated to developing standardised systems to measure the level of confidence in an effect estimate (quality of evidence or uncertainty). For instance, the GRADE methodology (Grading of Recommendations, Assessment, Development and Evaluation) is broadly used to evaluate the quality of evidence and the strength of recommendations derived from the evidence [34]. Under this framework, quality of evidence is classified into one of four levels (high, moderate, low, very low) based on an evaluation of aspects

including study limitations, consistency of results, indirectness of evidence, imprecision, and reporting bias.

When it came to standardising the communication of quality of evidence from risk assessments, a challenge was the need to take into account a broad variety of scientific disciplines with different standards for assessing evidence quality. It was not possible to impose standards from evidence-based medicine, as criteria for evaluating evidence quality could not be applied across disciplines (e.g., randomised controlled trials with human participants are not feasible when a substance has shown toxic effects in animal studies). In consultation with risk assessors, it became apparent that no unifying standardized framework existed to categorise quality of evidence across disciplines. Our solution to this issue was to incorporate a contextual explanation alongside ratings of 'low', 'moderate' or 'high' quality of evidence, containing information on the type of study conducted (e.g., case study, experiment), the quality of its design, the object of study (e.g. cell cultures, animals, humans), the number of participants (relative to the norms within a discipline: a few or a large number of participants), and consistency of results (see Fig 5 for an example). Nevertheless, future work to categorise evidence quality across scientific disciplines would be beneficial for standardising the communication of this type of evidence.

## Communicating dose-response relationships

Health effects that may result from toxic substances often depend on the dosage and duration of exposure. Consequently, the likelihood of experiencing health effects and the type or seriousness of an outcome can change substantially given different levels or duration of exposure. In the present study, we focused on the communication of the health-based guidance value which describes an exposure threshold under which no health effects are expected as long as individuals stay above or below this threshold (e.g., tolerable daily intake). However, we did not find any existing studies that evaluated formats for communicating these kinds of dose-response relationships.

A potential communication strategy that could be explored is the use of analogies to facilitate understanding of dose-response relationships. Analogies have been used in public risk communications by risk assessment agencies before, including the BfR, but their effect on comprehension has not been empirically tested. For instance, in addition to describing exposure scenarios using terms like "milligrams per kilogram bodyweight (mg/kg bw)", concrete everyday scenarios could be provided: "For healthy adults, the intake of up to 200mg caffeine within a short period of time is considered harmless to health. This corresponds to about two cups of filter coffee (200ml cups). Over the course of one day, twice this amount (i.e. 400mg) is considered to be safe in the general healthy population. This corresponds to about four to five cups of filter coffee" [41]. Analogies have been studied in the context of communicating about the diagnosticity of medical tests [42], improving comprehension of medical problems, and may be a promising communication strategy for communicating about dose-response relationships.

## Limitations

Due to the lack of existing risk communication tools or visualisation methods that could serve as examples for the redesign of the 2013 Risk Profile, the literature search was adapted to look in-depth at the communication of specific risk characteristics or risk communication recommendations in general. Our focus on conducting rapid reviews meant that we may have missed relevant material by restricting our searches (e.g., search time period, number of databases, study design, restriction of the inclusion to English and German papers). We also focused on

identifying systematic reviews and randomized controlled experiments and may have there-fore overlooked evidence in areas where no systematic reviews and RCTs could be identified (e.g., how to communicate quality of evidence). Further, the rapid reviews were not pre-registered.

Another limitation is that efforts to recruit people from disadvantaged groups for the focus group interviews turned out to be quite difficult. Due to the difficulties in accessing vulnerable groups, it remains unclear how broadly our results can be generalised to different disadvan-taged groups. Further, the focus group transcripts were not returned to participants in any group for comment or correction and the data were only analysed independently by one researcher and one student assistant for one interview. The V2 prototype was not assessed with risk managers or participants from the general public; only with risk assessors in the form of usability tests where only minor changes to the prototype V2 were made. Nevertheless, we are evaluating the effect of the newly developed final Risik Profile (2020 Risk Profile) on improving comprehension of risk assessment results in a randomized trial with members of the general public.

It is unclear how well the insights from the present study would transfer to other institu-tions engaged in risk communication, despite our efforts to involve individuals from different departments of the BfR, and different relevant target user groups. Applicability to other institu-tions involved in consumer health protection and food safety would need to be assessed. Fur-ther, while we aimed to improve risk assessment communication over a broad range of topics, certain topics may not have been covered sufficiently. For instance, the topics we examined may not have been able to measure how people evaluate the probability of occurrence depend-ing on the severity (mild vs. severe health consequences), the probability of contact (for instance, high consumption vs. low consumption) or differences in the quality of evidence (low uncertainty vs. high uncertainty). These questions warrant further research in future studies.

## Conclusion

We redesigned the 2013 Risk Profile to add content and improve the communication of the results of risk assessments to the general public, drawing on rapid literature reviews on the best evidence for communicating key characteristics of risk assessments, as well as on feedback from expert and non-expert groups. Usability tests with risk assessors from the BfR confirmed that while the redesigned risk profile contains more detailed information than the 2013 Risk Profile, it offered risk assessors the opportunity to present a more differentiated portrayal of the risks they assess. Remaining challenges include how to communicate dose-response rela-tionships and standardise quality of evidence ratings across disciplines. We offer suggestions on how to address some of these open issues, such as using analogies to simplify dose-response relationships or standardising classification systems for quality of evidence to promote consis-tency in ratings across topics. Future research could explore how best to address these chal-lenges as they relate to risk assessment communications, and to explore potential dissemination strategies for the risk profile across different channels targeting broader audi-ences. We are evaluating the effect of the 2020 Risk Profile on improving comprehension of risk assessment results in a randomized trial with members of the general public. [43].

## Supporting information

**S1 File. Literature search (short report)—Supplement material.**
(PDF)

**S2 File. Standards for Reporting Qualitative Research (SRQR).**
(PDF)

**S3 File. Focus group interviews—Supplement material.**
(PDF)

## Acknowledgments

We thank Maria Mildner, Alexandra Roth, Julia Beckhaus, Nicole Engelhardt and Ana Tomova for supporting the literature review process and focus group transcription and coding.

## Author Contributions

**Conceptualization:** C. Ellermann, M. McDowell, C. O. Schirren, A.-K. Lindemann, S. Koch, M. Lohmann, M. A. Jenny.

**Data curation:** C. Ellermann.

**Formal analysis:** C. Ellermann, C. O. Schirren.

**Investigation:** C. Ellermann, C. O. Schirren, A.-K. Lindemann.

**Methodology:** C. Ellermann, M. McDowell, C. O. Schirren, A.-K. Lindemann, S. Koch, M. Lohmann, M. A. Jenny.

**Project administration:** M. McDowell, A.-K. Lindemann.

**Supervision:** M. McDowell, M. Lohmann, M. A. Jenny.

**Writing – original draft:** C. Ellermann, M. McDowell.

**Writing – review & editing:** C. Ellermann, M. McDowell, C. O. Schirren, A.-K. Lindemann, S. Koch, M. Lohmann, M. A. Jenny.

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
