## [Editor Report · Decision Letter 0]

19 Aug 2021

PONE-D-21-14725

Identifying content to improve risk assessment communications within the Risk Profile: Rapid literature reviews and Focus groups with expert and non-expert stakeholders

PLOS ONE

Dear Dr. Ellermann,

Thank you for submitting your manuscript to PLOS ONE. After careful consideration, we have decided that your manuscript does not meet our criteria for publication and must therefore be rejected.

Specifically, the methods does not satisfy the standards of PLOS ONE.

I am sorry that we cannot be more positive on this occasion, but hope that you appreciate the reasons for this decision.

Yours sincerely,

Tim Mathes

Academic Editor

PLOS ONE

Additional Editor Comments (if provided):

Unfortunately, we cannot consider the manuscript for publication because the methodology does not satisfy the methodological standards of PLOS ONE. The main methodological concerns are:

- The approach or methods for initial search step are not described.

- I is not clear were the systematic review process has been rapid, i.e. what and where shortcuts were made, and what is the rational.

- It is quite unusual to synthesizes secondary (systematic reviews) and primary studies (RCTs), unless existing systematic reviews are out-dated (e.g. with new RCTs) and the primary studies are only used to supplement the systematic reviews.

- The literature search is technical flawed. Relevant MeSH-terms and keywords are missing.

- AMSTAR (1) was used although AMSTAR 2 already existed at the time of the assessment. In addition, it is not adequate to calculate an overall score for AMSTAR.

- Only experts from one institution were included in the focus groups.

- The sampling strategy and sample characteristics of the population are not described and thus it cannot be assessed if the sample is representative.
---

## [Author Response · Author response to Decision Letter 0]

4 Nov 2021

We thank the Editor Dr. Tim Mathes for taking the time to review a previous version of our manuscript. On review of the comments, we note that many of these comments can be easily addressed or the information can be found within the manuscript or in the Supplementary material. We provide responses to each comment below, having addressed each of these points.

- Editor: The approach or methods for initial search step are not described.

Our initial search step involved searching for existing examples of risk communication tools/strategies used by other risk assessment institutions. We did not include much detail in the manuscript as the search turned up only one other example (described in the manuscript) and we therefore elected to follow the approach used to develop evidence-based health information: focus the strategy on evidence for relevant endpoints and obtain information on the needs and requirements from target groups to develop prototypes. Nevertheless, we have provided more detail on the initial search for existing examples and the approach that was taken in the Methods.

“A search for the risk communication strategies of risk assessment institutions in Europe, as well as international and European risk assessment agencies (e.g., European Food Safety Authority, The European Food Information Council) found that almost all institutions used scientific articles or reports to publish their risk assessments, and none used a standardized approach to communicate results of individual risk assessments to lay audiences (with some exceptions for specific risk assessments of public interest [5] or general information or broad overviews of risk assessments [21]).”

- Editor: I is not clear were the systematic review process has been rapid, i.e. what and where shortcuts were made, and what is the rational.

In the methods, we describe in detail our search procedure and the way in which we defined our rapid review strategy: " Like systematic reviews, rapid reviews are part of the knowledge synthesis family [23] where “processes are accelerated and methods are streamlined to complete the review more quickly than is the case for typical systematic reviews” [24](p3). Rapid reviews are employed to provide an overview of the latest evidence on a specific topic to support decision-making while ensuring that the scientific imperative of methodological rigor is met [23, 24]. At the time we conducted the rapid reviews, there was no uniform method for conducting rapid reviews. Nevertheless, the methodological steps we took adhere to all recommendations outlined in the recently published Cochrane Rapid Reviews Methods group guidelines for conducting rapid reviews (with the single exception that the protocol was not published online) [23]. We describe our methods in detail in the Supplementary Material (see S1 File).”

Specifically, we restricted our search in the following way:

· Used a limited number of databases between 2 and 4, which varied according to the search (databases used were for instance: PubMed, PsycINFO, Scopus and Web of Science),

· Restricted the types of studies (English-language, last 5 or 10 years only for most of the searches),

· Limited search for grey literature to a search of reference lists and relevant journals,

· Limitation of double-checking during study selection procedure (30% of a random sample was double-checked).

We note potential limitations to the rapid review in the limitation section: e.g., that we may have missed relevant material by restricting our search (e.g., by search time period, number of databases, study design) and focusing only on high quality studies (e.g., systematic reviews, RCTs).

- Editor: It is quite unusual to synthesizes secondary (systematic reviews) and primary studies (RCTs), unless existing systematic reviews are out-dated (e.g. with new RCTs) and the primary studies are only used to supplement the systematic reviews.

We agree with the editor’s comment and would like to clarify that our approach is consistent with this strategy. First, rapid reviews served to generate an overview of existing approaches to communicating relevant aspects of risk assessment results to generate ideas for the revision of the risk profile.

We would like to emphasise that the purpose of these reviews was not to integrate findings into a single review; rather the searches were independent and focused on identifying communication strategies for each of the individual dimensions (e.g., communicating uncertainty, visualizing threshholds). This is consistent with an evidence-based approach and was also applied in the process of creating the German guideline on evidence-based patient information (Lühnen, J., Albrecht, M., Hanßen, K., Hildebrandt, J., & Steckelberg, A. (2015). Guideline for the development of evidence-based patient information: insights into the methods and implementation of evidence-based health information. Zeitschrift für Evidenz, Fortbildung und Qualität im Gesundheitswesen, 109(2), 159-165). In this regard, a combined search strategy was far too complex and our initial attempt to combine the search strategies led to an unmanageable number of hits with very inappropriate results. Therefore, the searches were split in order to be able to search in a more targeted way. Therefore, while one search summarizes evidence based on individual studies because there is not much literature on the topic (search on "How to communicate the severity of health impairments), others were based on findings from systematic reviews where there is a broader evidence basis. The findings from the individual searches were summarized for the purpose of informing tool development.

In Supplementary material S1, we describe that our focus was on systematic reviews, with or without meta-analyses, and primary studies, with the focus initially on systematic reviews for each topic. We would like to clarify that systematic reviews were prioritised, with RCTs supporting or supplementing these reviews where they were outdated or were no Systematic Reviews were available. Only for the uncertainty search and the search on "How to communicate the severity of health impairments" no systematic reviews could be identified and primary studies respectively, primary studies with findings from literature reviews were used.

We added this detail to the Methods and to the Supplementary material.

- Editor: The literature search is technical flawed. Relevant MeSH-terms and keywords are missing.

We included the following additional detail on how we identified search terms in the S1 supplement file to provide clarity for the readers: “We took a comprehensive approach to identify and revise relevant search terms. First, we dedicated multiple joint research meetings to identify relevant search terms and further refined these in several meetings with a dedicated information specialist from the library of the Max Planck institute for Human Development. These terms were iteratively adjusted to refine the search strategies multiple times. The final search terms excluded a number of initially promising terms because the hits were completely off topic and drastically increased the proportion of irrelevant studies (these terms were excluded after repeated reviews of search result samples).”

- Editor: AMSTAR (1) was used although AMSTAR 2 already existed at the time of the assessment. In addition, it is not adequate to calculate an overall score for AMSTAR.

The editor is correct that AMSTAR 2 already existed at the time of our assessment, and we made a conscious decision not to use AMSTAR 2 because, unlike AMSTAR 1, AMSTAR 2 was not validated at this time. This is also mentioned in the current Cochrane manual "Evaluation of systematic reviews Version 1.0" (Cochrane Deutschland, Arbeitsgemeinschaft der Wissenschaftlichen Medizinischen Fachgesellschaften – Institut für Medizinisches Wissensmanagement. „Bewertung von systematischen Übersichtsarbeiten: ein Manual für die Leitlinienerstellung“. 1. Auflage 2017. Available: Cochrane Deutschland: http://www.cochrane.de/de/review-bewertung-manual). We added information for our rationale for using AMSTAR 1 to the methods.

It is also correct that a sum score for an overall assessment at AMSTAR 1 was not intended in the first AMSTAR development paper. However, at the time the literature searches were carried out, this was still being discussed, as a sum score was also reported in the validation paper of AMSTAR 1. We therefore decided to report one at that time. In order to take into account the recent consensus on this, we would add statements about the individual limitations of the included systematic reviews in the supplement S1 instead of an overall assessment, if necessary.

- Editor: Only experts from one institution were included in the focus groups.

As described in our manuscript, several focus group interviews were conducted with different user groups. In addition to the literature search for identifying key elements for communicating risk communication aspects, initially focus group interviews were conducted with risk assessors who are solely responsible for preparing risk assessments and risk profiles in Germany. The aim was to identify barriers to communicating risk assessments, gather feedback on the 2013 risk profile, and develop ideas for a new risk profile from those who are directly involved in the risk assessment communication process.

The results of the literature reviews and focus group interviews with risk assessors led to the development of several prototypes for each dimension of the risk profile (V1). Subsequently, focus groups were conducted with different target audiences (risk managers from various fields of risk management, and the general public) to identify information needs and evaluate the prototypes for each dimension of the risk profile. The results of the focus groups with risk managers and lay people were also incorporated into the development of a risk profile redesign. The redesign of the risk profile was then tested in a user testing with risk assessors from the first focus group interviews, as they are the ones who have to complete it in the end.

- Editor: The sampling strategy and sample characteristics of the population are not described and thus it cannot be assessed if the sample is representative.

We conducted focus groups with three distinct user groups: national risk assessors, risk managers, and the general public. The BfR (German Federal Institute for Risk Assessment) has the statutory task in Germany to assess and communicate risk assessment results. They served as the target user-group and the profile design was developed with the goal to be implemented in this institution. As such, the participants from this institution were the relevant target and user-group, and we made sure to include a diverse sample of individuals from different departments to ensure diversity of opinions, needs, and feedback. We also conducted focus groups with other potential user groups, including risk managers (from different institutions) and members of the general public.

We addressed the editors comment in our limitation section, and make explicit that the transferability of the risk profile for use in other institutions should be assessed in future research. We also already address other potential limitations to our samples in the discussion.

Regarding the recruitment process, we provide details for each of the focus groups samples. For the risk assessors and risk managers, owing to the small number of participants within specific institutions, we could not provide detailed socio-demographic information while still ensuring data protection. Nevertheless, we have endeavored to include additional detail about the composition of the sample where possible:

For the focus groups with risk assessors, we report on page 7: "Individuals from all scientific departments at the BfR were invited to participate on a voluntary basis. The focus groups included participants with and without previous experience working with the 2013 Risk Profile. Participants were primarily risk assessors from various departments of the BfR (n=13), and included members of the press and legal units who were involved in communicating about risk assessments or the legal aspects of BfR risk communications (n=2). Due to the small number of participants, differentiated information on the fields of work and demographics is not reported in order to ensure data protection.”

For the risk managers, we report on page 7: "Participants (n =8) from the field of risk management were recruited from German ministries and federal offices by the BfR. Risk managers from these agencies use the results of risk assessments from the BfR to make decisions about risk management for consumers (e.g., determine limit values for certain substances in food; withdraw products from the market). Due to the small number of participants, differentiated information on the fields of work and demographics is not reported in order to ensure data protection."

We will add the information about the gender and the rough affiliation of the participants from the field of risk management, as well as the recruitment process: “Three men and five women were recruited, with five participants from the BVL (Federal Office of Consumer Protection and Food Safety) and three participants from the BMEL (Federal Ministry of Food and Agriculture). Individuals were nominated by the two institutions. BfR had sent a letter to the presidents of the two partner institutions describing the project and the objective of the focus groups and requesting that the organizations nominate individuals from as many different fields of work as possible who regularly come into contact with risk assessments / risk profiles. After receiving the contact details of suitable persons, information for the appointment was sent. A selection was thus made by the two partner authorities.”

For the general public, we report on pages 7-8: "The participants were recruited by an external market research company with the goal to recruit individuals from various backgrounds (e.g., age, education)… In the first group (n=8), the majority of participants had completed vocational training (n=5) and their mother tongue was German (n=7). The mean age was 44 years (age range 26-58 years) with equal gender representation. In the second group (n=8), three participants had completed vocational training, four had completed their studies (master or bachelor), one participant was currently studying and most reported German as their mother tongue (n=7). The mean age was 43 years (age range 23-62 years) and two participants were women.”

---

## [Decision Letter · Decision Letter 1]

22 Dec 2021

PONE-D-21-14725R1Identifying content to improve risk assessment communications within the Risk Profile: Literature reviews and Focus groups with expert and non-expert stakeholdersPLOS ONE

Dear Dr. Ellermann,

Thank you for submitting your manuscript to PLOS ONE. After careful consideration, we feel that it has merit but does not fully meet PLOS ONE’s publication criteria as it currently stands. Therefore, we invite you to submit a revised version of the manuscript that addresses the points raised during the review process.

We look forward to receiving your revised manuscript.

Kind regards,

Tim Mathes

Academic Editor

PLOS ONE

Journal Requirements:

Additional Editor Comments (if provided):

Reviewers' comments:

Reviewer's Responses to Questions

**Comments to the Author**

1. If the authors have adequately addressed your comments raised in a previous round of review and you feel that this manuscript is now acceptable for publication, you may indicate that here to bypass the “Comments to the Author” section, enter your conflict of interest statement in the “Confidential to Editor” section, and submit your "Accept" recommendation.

Reviewer #1: (No Response)

Reviewer #2: (No Response)

Reviewer #3: (No Response)

2. Is the manuscript technically sound, and do the data support the conclusions?

Reviewer #1: Yes

Reviewer #2: Yes

Reviewer #3: Yes

3. Has the statistical analysis been performed appropriately and rigorously? 

Reviewer #1: N/A

Reviewer #2: N/A

Reviewer #3: N/A

4. Have the authors made all data underlying the findings in their manuscript fully available?

Reviewer #1: (No Response)

Reviewer #2: No

Reviewer #3: Yes

5. Is the manuscript presented in an intelligible fashion and written in standard English?

Reviewer #1: Yes

Reviewer #2: Yes

Reviewer #3: Yes

6. Review Comments to the Author

Reviewer #1: The paper addresses a specific case related to BfR's role and work in Germany in the continuous development of comprehensible risk communication. The background, as inferred by the paper, is that different target groups involved in the risk communication chain have different views on importance, content, and outcome of the communication.

In this paper qualitative focus group methodology has been used to gather in-depth insights into aspects of the Risk Profile that may improve risk messaging. As such qualitative methodology does not provide generalisable results or knowledge of how a topic is understood in the general or target population. However, by taking certain precautions in the recruitment of respondents, valuable input and insight can be provided.

The paper addresses these issues in a satisfactory way.

I have one question: Is the qualitative evaluation of the 2013 Risk Profile conducted in 2015 published somewhere?

Reviewer #2: This is an interesting paper with the aim of improving the presentation methods for risk profiles on foods and consumer products as provided by risk assessment agencies. I believe this is an important area of development given the target audiences of these assessments and the increasing interest in science communication. The methods used are appropriate and generally well reported. It was a pleasure to review this manuscript. I have a few specific comments on the manuscript, which I hope are helpful.

General note:

Some parts of the manuscript seem unnecessarily lengthy in my opinion, e.g. the explanation of a PICO in S1 (section 2.1) and of the search techniques (use of thesaurus, key words, Boolean Operators). I don’t think it is necessary to describe standard review methods as long as it is reported in a transparent and reproducible way what was specifically done in the given review (e.g. reporting of the PICO elements, databases searched, search strategy etc.). I think it would make the paper more digestible to abbreviate such passages, but I understand that such choices can be deliberate depending on the audience.

Background:

In line 87/88 you refer to “EFSA explains” factsheets targeted at consumer audiences. This seems contradictory to step 1 (lines 209/10), where you state that searches for risk communication strategies used by international agencies did not yield any applicable results. Furthermore, you mention that there were some exceptions to this (line 211). It would be interesting to know whether and how these were considered or – if not – why they were not deemed relevant. Lastly, were there any relevant differences between the 2013 BfR version of the Risk Profile and the adapted Swedish “risk thermometer” that may have been relevant to advancing the BfR tool?

In line 94, you mention formats to disseminate findings from systematic reviews to non-expert audiences such as plain language summaries. However, it appears that reference 9 and 10 refer to studies conducted with health professionals or researchers. This seems misleading and I believe that there are better suited references, e.g. doi:10.1016/j.jclinepi.2014.04.009.

Methods/Results:

Line 235: I suggest rephrasing this. As I understand, you searched systematic reviews and trials on the effects of different communication strategies, not “current best practice recommendations” (e.g. as provided here: 10.1136/bmjopen-2019-036348). Also check line 400 on this.

Lines 247 ff: It should be mentioned here that not all the listed databases were searched for all key questions.

In the main manuscript it would be useful to state whether the focus group members received the Risk Profile beforehand. I did not find this information.

The final version of the Risk Profile on magnesium is an example with a verbal presentation of risks. It would be interesting to see an example where adverse effects are presented numerically.

Discussion:

The limitations section is relatively short and could address more issues. One limitation, for example, is that prototype V2 was not reassessed with risk managers and the members of the public. Thus, I am not sure the process is truly iterative. A second round could have highlighted additional information – for example (I am just thinking loudly here), how people feel about the presentation of the likelihood of occurrence for a low intake (vs. high intake or presentation of both).

Also, the interview guide provided in S3 seems ambitious for 1.5 to 2h focus groups. It would be interesting to reflect on this and whether there were topics that could not be addressed due to time constraints, especially among members of the public not familiar with the topic.

One important aspect of communication is the distribution of the information. This receives little attention in the manuscript. It would be interesting to discuss this aspect. To name one example, social media and mobile devices are major channels for infographics, but have specific requirements (size constraints, limited attention, “mobile first” etc.). Thus, the Risk Profiles would likely need to be modified for such purposes. The discussion provides an opportunity to highlight this limitation and opportunity for future work.

Supplement 1:

The numbers in Figure 1 deviate from the numbers in the preceding text (e.g. figure 1 reports 4 additional records identified from gray literature, whereas the text passage mentions three records identified through Google Scholar, 406 at ti/ab stage in figure vs. 405 in text and so on). It also seems odd that there are duplicates, even though only one database was searched. Have you checked that this was not an error from the automated deduplication in EndNote?

There also seem to be errors in the other flow charts in S1 (e.g. page 15: 161, 66, 76 and 93 records equate to 396, while the figure reports 391 initial records). Please make sure the numbers throughout the manuscript and supplements are correct and consistent.

Minor:

Line 90: should it say “to health”?

Line 92: the semicolon after Cochrane Collaboration seems to be superfluous

Line 137: I think the comma after “managers” is not needed

Line 175: “initial first step” seems tautologic

Line 225: consider removing “all”

References: Please check the references for correct citation style. There seem to be some errors here, e.g. reference 7 (publisher missing) and 14 (first names of authors written out, “and” between authors).

Reviewer #3: I was glad to have the opportunity to review this manuscript, as risk communication research as a whole does need to be improved starting actually from the communication of risk assessment process and results, that to be effective, needs to be crafted according to several factors and this paper well shows this complexity. The very first step to have food risks appropriately perceived and managed – both by risk manages and consumers – is to translate the output of the risk assessment into relevant, understandable, reliable, clear and possibly “operational” information/instruction to face that risk.

Therefore, I warmly recommend the publication of this work; as well I encourage the authors to conduct additional research to fix the underlined criticalities, as stated in the Discussion session (line 406). This is an important result that emerges from this work.

In addition, this work (the risk profile tool) has the potential to be adopted by the wider community of risk communication practitioners, and serve as a tools for example to make comparisons between countries in terms of use, understanding and increasing of risk communication efficacy to both risk managers and consumers.

Before publication, I would suggest some little changes. The paper is well written but I think that some little improvements could be done

- Line 97: a general definition of risk profile should be given (what is it? What is it meant for? What content/information should it deliver? Who prepares a risk profile?...); I understand that is resembles the description of the BfR risk profile and that this can be inferred from the text, but it is better to provide the reader with a general / ideal one and this study helped to find the best working one so far. See for example section 3.3 in EFSA’s Technical assistance in the field of risk communication https://efsa.onlinelibrary.wiley.com/doi/pdf/10.2903/j.efsa.2021.6574

- Line 176: how did you perform this task? Keywords used, websites/search engines searched, …

- Line 199: you mention here the “user-guide on how to complete the risk profile”: did you produce it as an output of Step 4? I don’t understand whether the “user-guide” is the risk profile template without information, simply the grid, or it is something different, e.g. a text that helps (guides) risk assessors to fill in the risk profile template with all the information needed?

- Line 311: step 3: which risk profile versions were discussed? Although your work is very detailed, it is difficult to seek for information through the main text and the supplementary materials, and the reader gets lost or does not easily remember each step of the methodology and the materials used at every given moment.

- Focus group results: did you consider creating a final table/figure to summarize focus group results to highlight common suggestions and discrepancies? I understand that the interview guides were different for each target audience, but the categorisation of results could help you draw a final map with major findings

7. PLOS authors have the option to publish the peer review history of their article (what does this mean?). If published, this will include your full peer review and any attached files.

Reviewer #1: No

Reviewer #2: **Yes:**

Reviewer #3: No

---

## [Author Response · Author response to Decision Letter 1]

16 Feb 2022

We thank the Editor Dr. Tim Mathes and the reviewers for the positive evaluation of our research, and for the many constructive suggestions to improve our manuscript. We have attempted to address all comments as we revised the manuscript. Our responses to the comments are provided in detail below.

We checked again the reference list and made sure all cited references are included.

We have checked our manuscript against these requirements.

We apologise that there may have been a misunderstanding. The study did not receive official funding. The study was financed by in-house funds of the two study partners and the contributions were contractually agreed between the two project partners.

“The study received financial support from the German Federal Institute for Risk Assessment (BfR) and the Max Planck Institute for Human Development (MPIB). The funding agreement ensures the authors’ independence in designing the studies, interpreting the data, writing, and publishing reports.

www.bfr.bund.de/en/home.html

www.mpib-berlin.mpg.de/en”

Not applicable

As stated in our previous submission, we are unable to share the transcripts from the focus group interviews as they contain potentially identifying or sensitive participant data. Participants were not informed or requested to allow for the data to be shared publicly. However, we have now uploaded the f4 analysis codebook from our qualitative analysis software to the Open Science Framework. The codebook details the coding of the transcripts and therefore represents the underlying data for our focus group analyses. We have modified the data availability statement to read:

“Relevant data are within the paper and its Supporting Information files. The data underlying the sub-study (focus group interviews) contain sensitive information and are protected by the Data Privacy Act. Thus, we are not able to share the transcripts from the focus group interviews. The study was approved by the Institutional Ethics Board of the Max Planck Institute for Human Development, Berlin, Germany. Interested researchers can contact the ethics committee of the Max Planck Institute for Human Development (Dr. Uwe Czienskowski, sciencec@mpibberlin.mpg.de).

A Codebook with details on the coding of the transcripts and number of text passages in the code per interview is available on the open science framework (https://osf.io/eqtd7/).”

As described above, we cannot publish the focus group transcripts for data privacy reasons. We therefore provide a clear process and contact information for our institutional ethics board for readers who would like further information. We also now include the data from our coding analysis on the Open Science Framework.

Reviewer #1:

The paper addresses a specific case related to BfR's role and work in Germany in the continuous development of comprehensible risk communication. The background, as inferred by the paper, is that different target groups involved in the risk communication chain have different views on importance, content, and outcome of the communication.

In this paper qualitative focus group methodology has been used to gather in-depth insights into aspects of the Risk Profile that may improve risk messaging. As such qualitative methodology does not provide generalisable results or knowledge of how a topic is understood in the general or target population. However, by taking certain precautions in the recruitment of respondents, valuable input and insight can be provided.

The paper addresses these issues in a satisfactory way.

We thank the reviewer for his positive evaluation of our study and the contributions of our manuscript.

I have one question: Is the qualitative evaluation of the 2013 Risk Profile conducted in 2015 published somewhere?

The report on the BfR evaluation of the 2013 Risk Profile is an internal report and unfortunately is not intended to be published by the BfR.

Reviewer #2:

This is an interesting paper with the aim of improving the presentation methods for risk profiles on foods and consumer products as provided by risk assessment agencies. I believe this is an important area of development given the target audiences of these assessments and the increasing interest in science communication. The methods used are appropriate and generally well reported. It was a pleasure to review this manuscript. I have a few specific comments on the manuscript, which I hope are helpful.

We thank the reviewer for his positive evaluation of our study and the contributions of our manuscript.

General note:

Some parts of the manuscript seem unnecessarily lengthy in my opinion, e.g. the explanation of a PICO in S1 (section 2.1) and of the search techniques (use of thesaurus, key words, Boolean Operators). I don’t think it is necessary to describe standard review methods as long as it is reported in a transparent and reproducible way what was specifically done in the given review (e.g. reporting of the PICO elements, databases searched, search strategy etc.). I think it would make the paper more digestible to abbreviate such passages, but I understand that such choices can be deliberate depending on the audience.

We thank the reviewer for this suggestion. We have now shortened some of these standard review paragraphs to make the paper more digestible for the reader.

Background:

In line 87/88 you refer to “EFSA explains” factsheets targeted at consumer audiences. This seems contradictory to step 1 (lines 209/10), where you state that searches for risk communication strategies used by international agencies did not yield any applicable results. Furthermore, you mention that there were some exceptions to this (line 211). It would be interesting to know whether and how these were considered or – if not – why they were not deemed relevant. Lastly, were there any relevant differences between the 2013 BfR version of the Risk Profile and the adapted Swedish “risk thermometer” that may have been relevant to advancing the BfR tool?

We appreciate the reviewer highlighting these apparent contradictions in our explanation of the initial search. The ‘EFSA explains’ factsheets do directly target consumer audiences and focus on providing easy to understand explanations for certain consumer-relevant topics. While some of these factsheets include visual elements and are structured to answer key risk questions, the factsheets are highly tailored to the specific risk assessment topic and do not provide a standardized approach that can be applied across risk assessment topics. Indeed, their structure and included visuals can differ quite markedly across topics. For these reasons, the EFSA explains factsheets were not considered as an existing standardized approach to assist our development of a standardized tool.

We have revised the manuscript in the Introduction, when we first mention the EFSA explains factsheets to highlight this difference:

“At the European level, risk assessments that are considered to be of particular interest to the public are communicated in formats targeted at more general audiences, such as the European Food Safety Authority’s “EFSA explains” Factsheets that develop tailored communications for specific topics (e.g., on caffeine, salmonella, or acrylamide in food; [5])”

We also include an explanation in the Method section where we describe the initial search method:

“The search revealed that almost all institutions used scientific articles or reports to publish their risk assessments. Further, none used a standardized approach to communicate results of individual risk assessments to lay audiences, with some exceptions. The European Food Safety Authority produces factsheets that are non-standardized, tailored communications for specific risk assessments of public interest [5] and general information on risk assessments or food safety topics in the form of infographics [6]).”

We also added some details on risk thermometer from the Swedish National Food Agency (NFA):

“The risk thermometer compares food-related risks by combining probability, severity and uncertainty into a single metric that is visualized in the form of a thermometer. The purpose of the tool is to enable direct comparisons of food risks according to this combined metric and does not provide an overview of the risk assessment topic in a format that is targeted at lay audiences.”

In line 94, you mention formats to disseminate findings from systematic reviews to non-expert audiences such as plain language summaries. However, it appears that reference 9 and 10 refer to studies conducted with health professionals or researchers. This seems misleading and I believe that there are better suited references, e.g. doi:10.1016/j.jclinepi.2014.04.009.

We thank the reviewer for drawing our attention to the more suitable reference. We have now cited it in the relevant section of the manuscript.

Methods/Results:

Line 235: I suggest rephrasing this. As I understand, you searched systematic reviews and trials on the effects of different communication strategies, not “current best practice recommendations” (e.g. as provided here: 10.1136/bmjopen-2019-036348). Also check line 400 on this.

Thank you for your comment, we have rephrased these sentences:

“Specifically, we reviewed literature on communicating the following key questions” (line 235)

and

“Further, for most of the five risk characteristics, there were no high quality studies (randomized controlled trials) or reviews for communicating results about the severity of health impairments, dose-response thresholds, or the uncertainty or quality of evidence.” (line 400)

Lines 247 ff: It should be mentioned here that not all the listed databases were searched for all key questions.

We have revised the sentence in line 249 to read:

“Owing to the different content within each topic, the databases were adjusted to the specific search such that the same set of databases were not used for all key questions. Specifically, for some searches, additional databases were employed to find relevant studies (e.g., the use of educational research databases to search for dose-response relationships.”

In the main manuscript it would be useful to state whether the focus group members received the Risk Profile beforehand. I did not find this information.

Only some of the risk assessors and risk managers were aware of the 2013 Risk Profile or had worked with it previously. As such, at the beginning of the session, risk assessors were asked for their opinion on the 2013 Risk Profile and its individual characteristics, as well as for ideas on how to improve the profile after presentation (line 290). We added the information that the 2013 Risk Profile was presented to risk assessors at the beginning of the focus group interviews.

“After presentation of the 2013 Risk Profile participants were asked for their opinions on the profile (see Fig 1) and its individual characteristics.” (line 304)

Participants from general population and risk managers were not shown the original 2013 Risk Profile but were instead shown two different prototypes of the risk profile (see Fig. 3) that were revised based on the literature review and feedback from the risk assessors interviews. These prototypes presented each key characteristic separately (i.e., participants did not receive a complete risk profile). In line 324 we wrote:

„Semi-structured focus group interviews were conducted with risk managers and members of the general public to elicit feedback on risk profile prototypes developed on the basis of Step (1) and (2), and to inform further revisions of the prototypes prior to user testing“

The final version of the Risk Profile on magnesium is an example with a verbal presentation of risks. It would be interesting to see an example where adverse effects are presented numerically.

Unfortunately, neither of the two risk assessment topics selected for the profile development contained numerical risks on adverse effects. These topics were chosen because they were everyday risks that would be relevant to the different user groups, and they presented different risk communication challenges across disciplines at the BfR (e.g., familiarity among consumers, communication of health based guidance value). Although we did not create a risk profile version that included numerical risks, there is extensive evidence from the risk communication literature on how to present numerical risks to facilitate understanding (e.g., absolute numbers or simple frequencies as opposed to relative risks). These recommendations are included in the user guide to inform how to present numbers within a risk profile (e.g., whenever possible, probability should be quantified with probabilities represented in absolute numbers: e.g., 3 out of 4 people who take more than [x mg] of [substance X] develop [endpoint]”).

We have now included this point in our Discussion section relating to quantifying probabilities in risk assessments, and refer to the extensive literature on presenting numerical probabilities to guide the reader to best practice recommendations:

“Indeed, although we did not include a risk assessment topic that presented risks numerically, there is an extensive literature on how to present probabilities in formats that facilitate understanding (e.g., absolute numbers rather than relative risks; including information on base rates or reference groups; Bonner et al. 2021).”

Discussion:

The limitations section is relatively short and could address more issues. One limitation, for example, is that prototype V2 was not reassessed with risk managers and the members of the public. Thus, I am not sure the process is truly iterative. A second round could have highlighted additional information – for example (I am just thinking loudly here), how people feel about the presentation of the likelihood of occurrence for a low intake (vs. high intake or presentation of both).

The reviewer is correct that we did not reassess the V2 prototype with risk managers and citizens, only with risk assessors. Our intention was to test the complete profile in a following step with a randomized controlled trial with members from the general public. We have included this point in the limitations section of the Discussion.

“The V2 prototype was not assessed with risk managers or participants from the general public; only with risk assessors in the form of usability tests where only minor changes to the prototype V2 were made. Nevertheless, we are evaluating the effect of the newly developed final Risk Profile (2020 Risk Profile) on improving comprehension of risk assessment results in a randomized trial with members of the general public.”

We also include a section in our Limitations to discuss the reviewer’s point about certain presentations of risks not being addressed in the current study:

“Further, while we aimed to improve risk assessment communication over a broad range of topics, certain topics may not have been covered sufficiently. For instance, the topics we examined may not have been able to measure how people evaluate the probability of occurrence depending on the severity (mild vs. severe health consequences), the probability of contact (for instance, high consumption vs. low consumption) or differences in the quality of evidence (low uncertainty vs. high uncertainty). These questions warrant further research in future studies.”

Also, the interview guide provided in S3 seems ambitious for 1.5 to 2h focus groups. It would be interesting to reflect on this and whether there were topics that could not be addressed due to time constraints, especially among members of the public not familiar with the topic.

The reviewer is correct that the interview transcript included a lot of questions for the focus groups. However, despite time constraints, we managed to ask all our intended questions within the timeframe, and sought to facilitate the groups so that all participants were able to contribute throughout the interviews. Nevertheless, given the many questions we had scheduled, it may have been the case that additional time dedicated to each question would have benefited the diversity of content we elicited from the group.

One important aspect of communication is the distribution of the information. This receives little attention in the manuscript. It would be interesting to discuss this aspect. To name one example, social media and mobile devices are major channels for infographics, but have specific requirements (size constraints, limited attention, “mobile first” etc.). Thus, the Risk Profiles would likely need to be modified for such purposes. The discussion provides an opportunity to highlight this limitation and opportunity for future work.

We thank the reviewer for pointing this out. The results of the risk assessments are published on the BfR website and serve as a basis for risk managers to make decisions or to communicate with consumers. In future, the risk profile could also be incorporated into direct communication to consumers (e.g., via the consumer protection centres’ website, facebook or twitter) or in various media reports dealing with the safety of certain foods or risks of consumers. We added this point for future research in the conclusion section.

Supplement 1:

The numbers in Figure 1 deviate from the numbers in the preceding text (e.g. figure 1 reports 4 additional records identified from gray literature, whereas the text passage mentions three records identified through Google Scholar, 406 at ti/ab stage in figure vs. 405 in text and so on). It also seems odd that there are duplicates, even though only one database was searched. Have you checked that this was not an error from the automated deduplication in EndNote?

We thank the reviewer for their attention to detail and picking up on these minor discrepancies in the numbers reported in our Supplementary material. We have rechecked our Endnote libraries and detected two errors due to the manual copying of studies during the screening process in Endnote: one study was excluded during the full text screening process but was inadvertently moved to the folder intended for papers excluded during title abstract screening; one study was erroneously included in triplicate). We have rechecked the remaining classifications to ensure no further errors in the Endnote classifications occurred.

There also seem to be errors in the other flow charts in S1 (e.g. page 15: 161, 66, 76 and 93 records equate to 396, while the figure reports 391 initial records). Please make sure the numbers throughout the manuscript and supplements are correct and consistent.

As stated above, we have rechecked the remaining Endnote classifications and flowcharts to ensure no further errors or inconsistencies occur in the manuscript and Supplementary Material. We have made the relevant corrections.

Minor:

Line 90: should it say “to health”?

Line 92: the semicolon after Cochrane Collaboration seems to be superfluous

Line 137: I think the comma after “managers” is not needed

Line 175: “initial first step” seems tautologic

Line 225: consider removing “all”

We have made each of the above mentioned suggested changes.

References: Please check the references for correct citation style. There seem to be some errors here, e.g. reference 7 (publisher missing) and 14 (first names of authors written out, “and” between authors).

We thank the reviewer for the note. We have corrected it.

Reviewer #3:

I was glad to have the opportunity to review this manuscript, as risk communication research as a whole does need to be improved starting actually from the communication of risk assessment process and results, that to be effective, needs to be crafted according to several factors and this paper well shows this complexity. The very first step to have food risks appropriately perceived and managed – both by risk manages and consumers – is to translate the output of the risk assessment into relevant, understandable, reliable, clear and possibly “operational” information/instruction to face that risk.

Therefore, I warmly recommend the publication of this work; as well I encourage the authors to conduct additional research to fix the underlined criticalities, as stated in the Discussion session (line 406). This is an important result that emerges from this work.

We thank the reviewer for the positive evaluation of our research. We are working towards further research on improving risk communication.

In addition, this work (the risk profile tool) has the potential to be adopted by the wider community of risk communication practitioners, and serve as a tools for example to make comparisons between countries in terms of use, understanding and increasing of risk communication efficacy to both risk managers and consumers.

Before publication, I would suggest some little changes. The paper is well written but I think that some little improvements could be done

- Line 97: a general definition of risk profile should be given (what is it? What is it meant for? What content/information should it deliver? Who prepares a risk profile?...); I understand that is resembles the description of the BfR risk profile and that this can be inferred from the text, but it is better to provide the reader with a general / ideal one and this study helped to find the best working one so far. See for example section 3.3 in EFSA’s Technical assistance in the field of risk communication https://efsa.onlinelibrary.wiley.com/doi/pdf/10.2903/j.efsa.2021.6574

We added some more details on the risk profile in line 107.

“Thus the Risk Profiles present a summary of a risk assessment in a format that makes the key criteria comprehensible at a glance. The Risk Profiles are developed at the time of the risk assessment in close collaboration with risk assessors and the communications department.

- Line 176: how did you perform this task? Keywords used, websites/search engines searched, …

We have added more information on how we performed this task to line 212).

“We searched the BfR website on European and international co-operations (www.bfr.bund.de/en/european_and_international_co_operations-10361.html) as well as search engines (e.g., Google) for risk assessment or risk communication institutions worldwide. In addition, experts from risk communication department of the BfR, who were involved in the 2013 Risk Profile development, were asked to share any other risk communication tools or visualization methods they were aware of.”

- Line 199: you mention here the “user-guide on how to complete the risk profile”: did you produce it as an output of Step 4? I don’t understand whether the “user-guide” is the risk profile template without information, simply the grid, or it is something different, e.g. a text that helps (guides) risk assessors to fill in the risk profile template with all the information needed?

The user guide was developed on the basis of the final risk profile (2020 Risk Profile). It serves to provide risk managers with guidance and detailed instructions on how to complete each section of the risk profile.

We added some details on the user guide to the methods and results section of step 4 (usability test and guide development).

Method section: "The user guide was drafted based on the V2 prototype and aimed to provide risk assessors with guidance on how to complete the risk profile (e.g., what information to provide in each section; best practices for presenting numerical risks etc). It was developed prior to the usability tests to assist risk assessors in completing a Risk Profile based on one of their recent risk assessment topics, and would be revised based on feedback from risk assessors during the usability tests.”

Results section: “Further, the draft user guide was developed in collaboration with risk assessors during the usability test. The purpose of the user guide was to provide risk managers with detailed instructions on how to complete each section of the risk profile, including guidance on what information to include and recommendations about what formats to present information to facilitate interpretation (e.g., to include numerical probabilities, if available, in absolute rather than relative numbers). These recommendations were also informed based on findings from the literature reviews (e.g., see Table 1). The draft user guide is currently being revised in coordination with working groups within the BfR.”

- Line 311: step 3: which risk profile versions were discussed? Although your work is very detailed, it is difficult to seek for information through the main text and the supplementary materials, and the reader gets lost or does not easily remember each step of the methodology and the materials used at every given moment.

We have revised Figure 2 to more clearly identify the various stages involved in the profile development, the insights gained at each stage, and the specific risk profile prototypes developed and/or shown to participants at each point in time. We have also now included more references to Figure 2 throughout the text to remind the reader of each stage. We hope this helps improve the reader’s orientation about the development process.

- Focus group results: did you consider creating a final table/figure to summarize focus group results to highlight common suggestions and discrepancies? I understand that the interview guides were different for each target audience, but the categorisation of results could help you draw a final map with major findings

As the reviewer suggests, as the results of focus group interviews were used to inform subsequent interviews and to develop and revise different prototypes, we did not summarise the results in a single table across all focus group studies. We have, however, attempted to summarise the changes to the prototype examples V1 from focus groups with risk managers and the general public in Table 3, as the interview guides and aim of focus group interviews were similar.

---

## [Decision Letter · Decision Letter 2]

29 Mar 2022

Identifying content to improve risk assessment communications within the Risk Profile: Literature reviews and Focus groups with expert and non-expert stakeholders

PONE-D-21-14725R2

Dear Dr. Ellermann,

We’re pleased to inform you that your manuscript has been judged scientifically suitable for publication and will be formally accepted for publication once it meets all outstanding technical requirements.

Kind regards,

Tim Mathes

Academic Editor

PLOS ONE

Additional Editor Comments (optional):

Reviewers' comments:

Reviewer's Responses to Questions

**Comments to the Author**

1. If the authors have adequately addressed your comments raised in a previous round of review and you feel that this manuscript is now acceptable for publication, you may indicate that here to bypass the “Comments to the Author” section, enter your conflict of interest statement in the “Confidential to Editor” section, and submit your "Accept" recommendation.

Reviewer #2: All comments have been addressed

Reviewer #3: (No Response)

2. Is the manuscript technically sound, and do the data support the conclusions?

Reviewer #2: Yes

Reviewer #3: Yes

3. Has the statistical analysis been performed appropriately and rigorously? 

Reviewer #2: N/A

Reviewer #3: N/A

4. Have the authors made all data underlying the findings in their manuscript fully available?

Reviewer #2: Yes

Reviewer #3: Yes

5. Is the manuscript presented in an intelligible fashion and written in standard English?

Reviewer #2: Yes

Reviewer #3: Yes

6. Review Comments to the Author

Reviewer #2: (No Response)

Reviewer #3: The peer review process definitely improved the quality of the manuscipt, that was already well written and reported a well executed study.

I definitely agree for publication.

I only have one remark, concerning Fig. 2

If I am correct, there should be 4 steps in the figures, as the reader can find 4 steps in the manuscript (both Methods and Results sections).

7. PLOS authors have the option to publish the peer review history of their article (what does this mean?). If published, this will include your full peer review and any attached files.

Reviewer #2: **Yes: **Roland Brian Büchter

Reviewer #3: **Yes: **Barbara Tiozzo

---

## [Editor Report · Acceptance letter]

1 Apr 2022

PONE-D-21-14725R2

Identifying content to improve risk assessment
communications within the Risk Profile: Literature reviews
and Focus groups with expert and non-expert stakeholders

Dear Dr. Ellermann:

I'm pleased to inform you that your manuscript has been deemed suitable for publication in PLOS ONE. Congratulations! Your manuscript is now with our production department.

Kind regards,

on behalf of

Dr. Tim Mathes

Academic Editor

PLOS ONE